# Cascading Risks for Preventable Infectious Diseases in Children and Adolescents during the 2022 Invasion of Ukraine

**DOI:** 10.3390/ijerph19127005

**Published:** 2022-06-08

**Authors:** Andrea Maggioni, Jose A. Gonzales-Zamora, Alessandra Maggioni, Lori Peek, Samantha A. McLaughlin, Ulrich von Both, Marieke Emonts, Zelde Espinel, James M. Shultz

**Affiliations:** 1Division of Pediatric Hospital Medicine and Global Health, Nicklaus Children’s Hospital, Miami, FL 33155, USA; andrea.maggioni@nicklaushealth.org; 2Division of Infectious Diseases, Department of Medicine, University of Miami Miller School of Medicine, Miami, FL 33136, USA; jxg1416@med.miami.edu; 3Division of Public Health Sciences, University of Miami Miller School of Medicine, Miami, FL 33136, USA; axm2161@miami.edu (A.M.); sam503@med.miami.edu (S.A.M.); 4Department of Sociology, Natural Hazards Center, and CONVERGE, University of Colorado Boulder, 483 UCB Boulder, Boulder, CO 80309, USA; lori.peek@colorado.edu; 5Pediatric Infectious Diseases Unit, Hauner Children’s Hospital, Ludwig-Maximilians-University, Lindwurmstrasse 4, 80337 Munich, Germany; ulrich.von.both@med.lmu.de; 6Department of Paediatric Immunology, Infectious Diseases and Allergy, Great North Children’s Hospital, Newcastle upon Tyne Hospitals NHS Foundation Trust, Queen Victoria Road, Newcastle upon Tyne NE1 4LP, UK; marieke.emonts@newcastle.ac.uk; 7Translational and Clinical Research Institute, Newcastle University, Queen Victoria Road, Newcastle upon Tyne NE1 4LP, UK; 8Sylvester Comprehensive Cancer Center, Department of Psychiatry and Behavioral Sciences, University of Miami Miller School of Medicine, 1121 NW 14th St., Miami, FL 33136, USA; z.espinel@miami.edu; 9Center for Disaster & Extreme Event Preparedness (DEEP Center), P3H: Protect & Promote Population Health in Complex Crises, Department of Public Health Sciences, University of Miami Miller School of Medicine, Don Soffer Clinical Research Center Room 1507, 1120 NW 14 St., Miami, FL 33136, USA

**Keywords:** war, invasion, displacement, refugee, Ukraine, children, infectious diseases, COVID-19, HIV, measles

## Abstract

Russia’s military incursion into Ukraine triggered the mass displacement of two-thirds of Ukrainian children and adolescents, creating a cascade of population health consequences and producing extraordinary challenges for monitoring and controlling preventable pediatric infectious diseases. From the onset of the war, infectious disease surveillance and healthcare systems were severely disrupted. Prior to the reestablishment of dependable infectious disease surveillance systems, and during the early months of the conflict, our international team of pediatricians, infectious disease specialists, and population health scientists assessed the health implications for child and adolescent populations. The invasion occurred just as the COVID-19 Omicron surge was peaking throughout Europe and Ukrainian children had not received COVID-19 vaccines. In addition, vaccine coverage for multiple vaccine-preventable diseases, most notably measles, was alarmingly low as Ukrainian children and adolescents were forced to migrate from their home communities, living precariously as internally displaced persons inside Ukraine or streaming into European border nations as refugees. The incursion created immediate impediments in accessing HIV treatment services, aimed at preventing serial transmission from HIV-positive persons to adolescent sexual or drug-injection partners and to prevent vertical transmission from HIV-positive pregnant women to their newborns. The war also led to new-onset, conflict-associated, preventable infectious diseases in children and adolescents. First, children and adolescents were at risk of wound infections from medical trauma sustained during bombardment and other acts of war. Second, young people were at risk of sexually transmitted infections resulting from sexual assault perpetrated by invading Russian military personnel on youth trapped in occupied territories or from sexual assault perpetrated on vulnerable youth attempting to migrate to safety. Given the cascading risks that Ukrainian children and adolescents faced in the early months of the war—and will likely continue to face—infectious disease specialists and pediatricians are using their international networks to assist refugee-receiving host nations to improve infectious disease screening and interventions.

## 1. Introduction

Russia’s invasion of Ukraine, the largest military assault in Europe since World War 2, commenced on 24 February 2022, producing disproportionate effects for Ukrainian children and adolescents. Among the complex population health challenges facing Ukrainian youth, this article focuses on the elevated threat of the transmission of preventable infectious diseases. Two-thirds of Ukrainian children were displaced from their homes during the volatile early months of the conflict, and risks regarding the spread of infectious diseases were dynamically transformed due to ongoing fighting, population movements, and several other factors.

This article describes intersecting patterns of preventable infectious disease risks for Ukrainian children and adolescents—and for European nations receiving Ukrainian refugee flows. We detail the multi-layered interactions of the conflict with infectious disease risks for Ukrainian youth, with a specific focus on COVID-19 transmission, vaccine-preventable diseases, HIV transmission, wound infections from conflict-associated medical trauma, and war-related sexually transmitted infections. We conclude by briefly discussing the challenges for and progress with the monitoring of communicable disease risks for Ukrainian refugees during the first months of the war.

## 2. Infectious Disease Spread in the Context of Invasion, Occupation, and Displacement

Patterns of infectious diseases must be considered from the perspective of Ukrainian children and adolescents on the move [1]. Russia’s campaign of destruction, mass displacement, and depopulation has produced conflict-induced population mobility on a scale not witnessed in Europe since the 1940s. As of 31 May 2022, 6.8 million Ukrainians had become refugees, leaving their homeland and seeking safety and sanctuary in other nations [2,3]. Another 8.0 million Ukrainians were internally displaced during this period, left living precariously inside Ukraine after being evacuating from homes that were destroyed or fleeing communities under assault and bombardment [3]. Taken together, 14.8 million Ukrainians—33.5% of the nation’s 44.1 million citizens—had become forced migrants within the span of three months. The Ukraine war alone inflated the worldwide tally of displaced persons from 84 million to 99 million, a 17% overall increase [4].

Almost 90% of Ukrainian migrants were women and children. Among Ukraine’s 7.5 million children under the age of 18, only one-third were able to remain in their home communities as of 31 May 2022. Of the remaining two-thirds, more than 2.2 million children had migrated to European countries and 3 million were internally displaced inside Ukraine [5].

During these early months of the war, Ukrainian youth experienced variable and rapidly changing living conditions related to invasion, occupation, transitory battlelines, widespread bombardment, ongoing displacement, and in some locales, return migration to their homes. The simmering conflict, dating from the 2014 annexation of Crimea, was swiftly amplified by the full-scale 2022 invasion and ultimately placed portions of eastern and southern Ukraine under Russian control. Three months into the conflict, Russian forces had withdrawn from territory seized in the north to concentrate on enlarging territorial gains in their offensive in the east and south of Ukraine.

The mechanisms for accurately assessing infectious disease transmission for Ukrainian children and adolescents rapidly deteriorated at the onset of the incursion. These gaps in health surveillance were especially dire for those living in conflict zones and Russian-occupied areas. Even for children remaining in their home communities in areas that were spared from the worst of the active fighting and from artillery and aerial bombardment, surveillance of infectious diseases was hampered by the primary focus nationwide on population survival and the care of wounded civilians while under attack. Moreover, millions of Ukrainian refugee children were widely scattered across receiving nations and there was no means of systematically tracking them or assessing their health status [5].

## 3. COVID-19 Transmission Risks for Ukrainian Children and Adolescents Affected by Conflict

The most ubiquitous preventable infectious disease risk for children and adolescents during the first months of Russian aggression was the ongoing COVID-19 pandemic. During the early months of the conflict, the highly transmissible SARS-CoV-2 Omicron variant was predominant throughout Europe and globally. In Ukraine, recorded COVID-19 cases peaked at 241,000 during the first week of February 2022 and declined to 111,000 cases one week prior to the Russian invasion [6]. According to World Health Organization (WHO) surveillance data, Ukraine was one of the most affected European countries, surpassing 5 million cumulative cases toward the end of April 2022 [7,8]. Following Russia’s assault, Ukraine’s COVID-19 surveillance reporting was severely disrupted.

From the moment of initial invasion, Ukraine experienced a sharp reduction in COVID-19 testing related to the closure of medical centers nationwide, whereas diagnostic laboratories were reduced to operating at a fraction of capacity [9]. Counts of symptomatic cases in Ukrainian children were unavailable or inaccurate, given the extraordinary levels of displacement of Ukrainian youth. Furthermore, because children develop mostly mild forms of the disease, COVID-19 illness is likely to be missed, a situation that has occurred not only in Ukraine, but also worldwide. Nevertheless, on rare occasions, Omicron has been demonstrated to cause severe cases of COVID-19 in children and atypical presentations such as MIS-C (multi-inflammatory syndrome in children) that can potentially lead to death [10].

Access to hospital-based treatment for severe COVID-19 cases, including both adults and children, rapidly deteriorated as hostilities escalated. In areas occupied by Russian troops and nearby communities under Ukrainian control that had been subjected to heavy bombardment, healthcare centers were damaged or destroyed and there was a critical shortage of medical personnel. Military and civilian war casualties received priority when allocating beds and services. Oxygen was in short supply.

Prior to the invasion, the Ukrainian population was highly susceptible to COVID-19 transmission due to low rates of vaccination. Mistrust and concerns about side effects may have played an important role in the people’s willingness to receive a vaccine. In addition, the unavailability of vaccines in certain locations and restrictions favoring predominantly high-risk groups could have been barriers that limited vaccination [11]. In Ukraine, only 35.7% of the population had received two doses of the vaccine by late February 2022 when the war began, a much lower proportion than the average reported for European Union nations (71%) [6,12]. Vaccination rates ranged from 65% in Kyiv, the capital of Ukraine, to as low as 20% in outlying regions [13]. Less than 2% of the Ukrainian population had received a third (“booster”) dose of vaccine, which confers a much higher level of protection against the highly transmissible Omicron variant and its still-more infectious BA.2 sub-lineage that was actively circulating [14]. Ukrainian children were largely unvaccinated for COVID-19 when hostilities began [8], and vaccines were never provided for children under 12. Although adolescents had finally become eligible for COVID-19 vaccination, the national campaign to vaccinate children over the age of 12, launched on 13 January 2022, lasted only six weeks before it was brought to a halt by the conflict [9].

Behavioral measures proven to effectively mitigate the spread of COVID-19 worldwide, including the use of masks, social distancing, and limiting the size of social gatherings, could not be effectively implemented for Ukrainian children or adults in wartime. Supplies of face masks were very limited nationwide, particularly in the areas of Russian occupation [15]. Further, the cessation of systematic COVID-19 surveillance was compounded by the impossibility of quarantining symptomatic people, conducting contact tracing, and isolating contacts [9].

Population mobility elevated risks of airborne disease transmission, particularly of COVID-19. Ukrainian youth on the move experienced a multi-step sequence of potential infectious disease exposures. During the first months of the conflict, Ukrainian citizens in cities throughout the nation experienced heavy bombardment, artillery fire, and fusillades of missiles. Many children spent nights—or more protracted periods of time—crowded into subways and durable structures that served as bomb shelters. Some were able to board trains and buses to take them to cities in Western Ukraine. Ultimately, many crossed into border countries where their first days involved long lines at crowded reception centers. Thereafter, these refugee children and their mothers were dispersed to communities throughout the receiving countries. Although some children were able to relocate to the homes of relatives or friends in Europe, most found temporary housing inside border nations or continued their refugee journey into more distant countries. At each point along this migratory trajectory, children congregated with others for varying periods of time, invariably sharing the airspace and the respiratory aerosols produced by people in close proximity.

After arriving in European host countries, Ukrainian child and adolescent refugees and their family members achieved a level of safety from war-related, acute threats to life and health [16]. However, they remained at elevated risk of contracting a range of communicable diseases, especially COVID-19, due to crowded living conditions and poor access to water, sanitation, housing, and healthcare [17]. Receiving countries such as Hungary, Poland, the Republic of Moldova, and Romania had support from the WHO to devise strategies to try to minimize the spread of COVID-19, with many European countries providing free COVID-19 vaccines and disease surveillance for arriving Ukrainian refugees.

## 4. Vaccine-Preventable Disease (VPD) Risks for Ukrainian Children and Adolescents

As described, COVID-19 transmission risks were overlaid upon the widespread susceptibility of Ukrainian children to a spectrum of serious vaccine-preventable diseases (VPDs). Risks of spreading VPDs were complicated by the mass displacement and migration of millions of Ukrainian children. Keenly aware of this complex situation, the European Center for Disease Control and Prevention (ECDC) released detailed operational guidance for European health authorities. This guidance specified the preventable infectious disease risks, particularly for women and children, associated with the influx of Ukrainian refugees [17]. The ECDC document outlined policy recommendations for minimizing infectious disease spread among displaced Ukrainians and the citizens of the receiving countries. Among VPDs, the ECDC indicated that monitoring poliomyelitis and measles in children crossing into Europe or displaced within Ukraine were top priorities.

Widespread vaccine hesitancy in Ukraine during the first decades of the 2000s contributed to very low childhood vaccination rates for the diphtheria-tetanus-pertussis (DTP) vaccine; the measles, mumps, and rubella (MMR) vaccine; and the oral polio vaccine [8,18]. In 2018, a 12,000 case measles outbreak erupted in Ukraine, triggering emergency vaccination [19]. When the fighting broke out in February 2022, the two-dose measles vaccination rate was just 81.9%. This percentage exceeds the vaccination rates achieved for COVID-19, but this is deceptive. This level of measles vaccination is inadequate to protect against rapid outbreak spread, given measles’ very high reproduction number (the average number of persons who will be infected by one person with the disease) and ease of transmissibility. This has produced a high-risk situation, particularly with most Ukrainian children on the move.

Inadequate vaccine coverage also created significant vulnerability to polio, particularly for children under the age of 6. Nationwide, Ukrainian polio vaccination coverage was 80% in 2021, ranging from as low as 60% to 99%, and prompting an urgent nationwide vaccination campaign that commenced on 1 February 2022, but was disrupted three weeks later by the Russian incursion [20].

Seasonal influenza was circulating when the Russian invasion began but only 167,000 Ukrainians (0.4% of the population) had been vaccinated [20].

Tuberculosis (TB) remains a critical public health concern in Ukraine, particularly because the rate of multi-drug-resistant tuberculosis (MDR-TB) is exceptionally high, accounting for 24–29% of newly diagnosed cases [20]. Furthermore, Ukraine ranks second among European nations in HIV/TB coinfection; just under one-quarter (22%) of Ukrainians living with HIV are coinfected with TB.

Reports that many Ukrainians were bringing their pets with them as they migrated raised concerns for rabies, which is endemic among sylvatic animals, cats, and dogs in Ukraine [21].

Ukrainian children arriving in several European host countries, including Poland, received medical evaluations, along with vaccination [22]. The WHO has worked with border nations that are receiving refugees, including Hungary, Romania, and the Republic of Moldova, to upgrade their disease surveillance systems and immunization services [23]. Many children lost their vaccination records when fleeing in response to hostilities, which has complicated efforts to ensure that children are fully immunized from a variety of infectious diseases.

## 5. HIV Risks in Ukrainian Children and Adolescents 

As a context for HIV risk in war-affected children and adolescents, Ukraine currently ranks second in Eastern Europe in terms of the prevalence of HIV, with an estimated 260,000 cases and 3000 to 5000 deaths in adults and adolescents annually [8]. Intravenous drug use and heterosexual transmission are prominent HIV exposure risks for both adults and adolescents.

HIV cases in Ukraine rose almost 10% annually from 2007 to 2017. Yet only 40% of HIV positive adults and 54% of HIV-positive children were receiving anti-retroviral treatment before the war [8]. Anti-retroviral treatment was strongly recommended to prevent vertical transmission from infected mothers to children; however, the rapidly deteriorating situation in Ukraine dramatically limited the access to HIV therapy in pregnant women with the subsequent risk of HIV infection in their newborns. The war also raised concern that other existing barriers to the prevention of mother-to-child transmission (PMTCT) services may become even more pronounced [24].

The war in Ukraine has wreaked havoc on the entire health system and led to the closure of more than 40 facilities that offered anti-retroviral therapy and prevention services, all of which can put the country at risk of HIV resurgence [25]. More than 100,000 people with HIV were living in areas directly affected by the war, with 59,000 on HIV treatment [26]. Less than 40% of these people were able to move outside the war zone, leaving a significant number of individuals in conditions of poor access to HIV care. The United States President’s Emergency Plan for AIDS Relief (PEPFAR) delivered more than 18 million doses of antiretrovirals to be distributed in partnership between the Ministry of Health’s Public Health Center and 100% Life, the largest organization for people living with HIV in Ukraine. Nevertheless, it was very challenging to reach persons with HIV located in occupied zones and front-line battle areas [25].

Meanwhile, European border nations that were receiving Ukrainian refugees did not have sufficient resources to ensure treatment for people living with HIV. Untreated HIV-infected refugees posed a risk for the serial transmission of HIV to other refugees and members of the general population of the host countries [27]. Aware of this situation, UNAIDS and the WHO actively worked with European Union countries to provide antiretroviral therapy for refugees [25]. 

## 6. Wound Infections from Conflict-Associated Medical Trauma

Among the most feared complications of war injuries are wound infections, which will likely continue to increase significantly in the Ukrainian civilian population, given the numerous attacks to buildings, schools, museums, and healthcare facilities [28]. Through May 2022, the UNHCR received reports of 415 children who had sustained injury in the war, a figure that was presumed to be a gross undercount [29]. Severe burn injuries in children have also been reported [30]. Furthermore, children have been subject to penetrating wounds, puncture wounds, fractures, and lacerations sustained during bombardment, in structural collapses, and while navigating hazardous debris and wreckage in war-torn portions of Ukraine.

Wounds can become easily infected, mainly by community acquired microorganisms, such as *Staphylococcus epidermidis* and *Bacillus* spp., pathogens that predominate in the first week of an injury [31]. Despite the relatively low virulence of these bacteria, their treatment in children can be very challenging, especially during wartime, which has further diminished access to antibiotics and limited the supply of wound care kits. These issues, coupled with low rates of DTP (diphtheria, tetanus, pertussis) vaccination, may complicate the course of a contaminated injury, potentially leading to fatal *Clostridium tetani* infections [32].

The wide availability of antibiotics has substantially reduced mortality from wound infections but has also propelled the emergence of antimicrobial resistance. This phenomenon has been observed in the context of war-related infections, particularly for patients requiring prolonged hospitalization. During the second week of an injury, Gram-negative pathogens become the predominant bacteria, with many of them harboring complex mechanisms of resistance [31].

Historically, there have been reports of multi-drug-resistant *Acinetobacter baumannii* infections associated with military operations in Iraq, and frequent cases of extended spectrum beta-lactamase (ESBL) producing Enterobacteriaceae isolated as colonizers in military medical centers from the US and Germany between 2005 and 2009 [33,34]. According to a study published in 2017, the colonization by methicillin-resistant *Staphylococcus aureus* (MRSA) in Swiss military medical centers was about 10 times higher than that described for the general Swiss population [35].

Recent reports have shown the emergence of highly resistant bacteria in Ukraine as well. According to the Central Asian and European Surveillance of Antimicrobial Resistance (CAESAR) network in 2020, the proportion of ceftriaxone-resistant *Escherichia coli* can be as high as 53%, with an alarming rate of carbapenem-resistant *Acinetobacter* spp. of 77% in hospitalized patients [36]. 

There is a valid concern for the development of resistant infections in children with underlying conditions receiving care in Ukrainian hospitals. Infection control programs have been difficult to maintain due to diminished resources and increased caseloads of war-associated medical trauma patients following repeated attacks on civilian populations. European nations that have received inflows of Ukrainian refugees adopted stringent healthcare infection control policies, including pre-emptive isolation and screening for multidrug-resistant Gram-negative organisms in individuals with a history of hospital admission in Ukraine within the past 12 months [37].

## 7. Sexually Transmitted Infection (STI) Risks for Ukrainian Children and Adolescents 

Historically, armed conflicts have been associated with sexual abuse and human trafficking. Women and children are particularly vulnerable. Rape has been used a weapon of war to systematically terrorize and displace populations. Presumptive incidents of gender-based violence (GBV) perpetrated on Ukrainian women and youth at the hands of invading Russian troops were reported within the first weeks of the invasion, with allegations proliferating during subsequent months of conflict and occupation [38]. Potentially preventable infectious diseases and sexually transmitted infections (STIs), including HIV, may result from GBV and the coercive sexual exploitation of youth.

Ukrainian youth, particularly unaccompanied adolescents and growing numbers of children who have been orphaned by the war, represent high-risk subpopulations for GBV while in congregated settings and while migrating. UNICEF expressed specific concerns regarding Ukrainian children separated from their families as easy targets for human trafficking and sexual abuse [39]. Countries accepting Ukrainian refugees were encouraged to screen for STIs, including HIV, with special attention to female adolescents and children. The high likelihood of GBV leading to STIs is supported by studies conducted with internally displaced women in conflict-affected portions of Ukraine in the aftermath of the 2014 Russian invasion and annexation of Crimea [40].

## 8. Challenges and Resources for Monitoring Communicable Disease Risks for Ukrainian Refugees

Providing consistent and comprehensive infectious disease screening, timely intervention, and medical care for Ukrainian child refugees has been challenging due to many compounding factors [41]. The scale of forced migration, with large numbers of refugees and IDPs on the move, settling temporarily, and relocating to other communities or countries, creates formidable obstacles for providing continuity of care, as well as for tracking and maintaining accurate medical records.

Another complication involves refugee families arriving in European countries that have been hosted in private homes. Private residences are outside the jurisdiction of community public health services that focus on refugees in shelters and public housing. Refugees who are accommodated in private homes may not receive screening for infectious diseases and other critical health services. Going forward, it will be critical to integrate refugee children and their caregivers into community pediatric services in a timely and holistic manner.

Challenges notwithstanding, the Ukrainian refugee crisis has triggered an astonishingly rapid, rich, and resourced response. Since the beginning of the war, neighboring European countries such as Poland, Romania, Hungary, Slovakia, the Republic of Moldova, and Lithuania have generously provided a safe haven for the massive refugee flows.

Arriving refugees have been predominantly women, children, and caregivers, with male family members remaining in Ukraine to defend the homeland. Refugees arrive with urgent needs for safety, security, and basic survival necessities. Border countries have opened a network of reception centers to provide migrants with shelter, food, healthcare, information, training, specialized interventions, and other essential services [42].

Children with complex medical conditions have been prioritized to receive care. Special attention has been given to continuing care for child cancer patients, including those with hematological conditions that may be complicated by resistant infection.

A broad spectrum of humanitarian actors, including many WHO and UN agencies, are on-scene, providing services both inside Ukraine and throughout European countries. Most pertinent for our focus on children is the work of UNICEF, playing a crucial role in providing support for refugee children and their families [43]. Inside Ukraine, UNICEF has distributed life-saving health and medical supplies for nearly 2.1 million people in war-affected areas, reaching more than 610,000 children and caregivers with mental health and psychosocial support services, and providing educational supplies to nearly 290,000 children.

In refugee-hosting countries, UNICEF is supporting national, municipal, and local systems that deliver essential services and protection, particularly for the most vulnerable children. This includes anti-trafficking training for border guards, expanding learning opportunities and integrating refugee children into schools, procuring vaccines and medical supplies, and establishing play and learning hubs that provide young children with a much-needed sense of normalcy and respite [43]. Almost 300,000 vulnerable families have registered for the UNICEF Ministry of Social Policy humanitarian cash assistance program [43].

UNICEF has also established Blue Dots—safe spaces situated along border crossings in border countries that provide refugee children and families with critical information and practical support to help them in their onward journeys [44]. Blue Dot staff identify and register children traveling on their own and connect them to protection services. Blue Dots offer referral services to women, including those who are survivors of GBV. For children, Blue Dot hubs provide a safe, welcoming space to rest, play, and benefit from structured activities and psychosocial support from trained staff [44]. To date, 25 Blue Dots have been established along major transit routes in Moldova, Romania, Poland, Italy, Bulgaria, and Slovakia [43,44].

At border crossings and transit sites, medical care for refugees is being provided by both local humanitarian groups and international organizations, such as the Red Cross [45]. For example, in Poland, there are as many as 450 Red Cross medics working at border sites [45]. The provision of basic first aid and psychosocial support is supplemented by referrals to higher levels of care provided by national healthcare systems as necessary.

Multiple European governments have introduced legislation to allow refugees to access healthcare in hosting countries. On 3 March 2022, European Union (EU) members agreed to activate the Temporary Protection Directive for Ukrainian refugees, granting them access to healthcare across the EU. At least 10,000 hospital beds across the EU have been secured for refugee patients [45].

Vaccination centers have been established in receiving countries to ensure the continuity of COVID-19 vaccinations and child immunizations for refugees. In the UK, a scheme has been devised to allow Ukrainian refugees to receive consultations and care through the National Health Service (NHS) [45].

Specific to infectious diseases, inside Ukraine, the Ministry of Health is working with local clinics and NGOS such as the Alliance for Public Health and 100% Life to facilitate the availability of antiretroviral therapies for people living with HIV who have being displaced from their usual sources of care. The Ukrainian government has activated hotlines to provide guidance to people living with HIV, TB, and viral hepatitis. HIV care for refugees has also been offered at no cost by several European nations, including Belgium, Bulgaria, Croatia, and Cyprus [46].

European national and international networks of pediatric infectious disease specialists have been fortified during the COVID-19 pandemic, along with clinical trial networks for pediatric TB and HIV. During the war, these established networks have facilitated close contacts with Ukrainian counterparts. Among the medical societies active in this process, the European Society for Paediatric Infectious Diseases (ESPID) and the European Academy of Paediatrics (EAP) have forged close linkages with the Royal College of Paediatrics and Child Health (RCPCH) in the UK to share critical health information regarding preventable infectious diseases and other health conditions for Ukrainian refugee children and adolescents [41,47,48,49,50].

## 9. Conclusions

Russia’s invasion has produced mass internal and cross-border displacement of Ukraine’s children and adolescents. Overlaid upon the ongoing COVID-19 pandemic, and complicated by low rates of childhood vaccination, these co-occurring events have produced a compound disaster scenario. The surveillance and control of preventable infectious diseases in children and adolescents was one of the most complex population health challenges generated by the conflict, with the potential for increasing the spread of disease throughout receiving countries in Europe and beyond.

Concerted efforts on the part of networks of infectious disease experts and pediatric clinicians have attempted to intervene to optimize health and prevent the spread of infectious disease among this vulnerable population of conflict-displaced Ukrainian youth. For example, The Ukraine Children’s Action Project was established to support displaced Ukrainian children’s health and educational needs. As the conflict continues, more children will be injured, displaced, and left at risk to the spread of infectious disease. These efforts are therefore vital for protecting the health and well-being of the youngest survivors of the war.

## Data Availability

Not applicable.

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
