# Peer review of "Cascading Risks for Preventable Infectious Diseases in Children and Adolescents during the 2022 Invasion of Ukraine"

_ijerph, 2022, doi:10.3390/ijerph19127005_

Round 1
Reviewer 1 Report
Dear authors
Thank you for this research paper it reveals the ugly truth behind wars. I find this research fascinating and it is the core of the study of public health.
Author Response
Reviewer #1:
Note to reviewer: This paper was developed in response to a direct invitation from guest editor, Dr. Bozzola, to first author, Dr. Andrea Maggioni, to contribute a paper for the special IJERPH issue on preventable infectious diseases in children. The Ukraine conflict provided a compelling tale to tell, and we directed our writing to this evolving situation that threathens the health and welfare of Ukrainian children, with primary emphasis on infectious diseases.
Thank you for your review.
Comment: Thank you for this research paper it reveals the ugly truth behind wars. I find this research fascinating and it is the core of the study of public health.
Response: Thank you very much for your comment. We concur.

Reviewer 2 Report
I have read the article by Maggioni et.al. with great interest. However, after thorough reading, I could not understand the aim of the manuscript. The authors provide commentary on the current situation in Ukraine, and how the recent war situation may lead to public health crisis in displaced children, in terms of infectious diseases, ultimately affecting the countries to which the migrant children may arrive to.
My biggest concern regarding this manuscript is that it does not qualify as an article, perhaps it is wiser to shorten the manuscript and use as a commentary or letter to the editor if feasible.
The manuscript is vague, does not provide sufficient evidence nor provide thorough discussion of the topic in question. Most importantly, the statistical data are not supported with proper referencing, and while the conclusions arrived to by the authors are logical and may in fact be the case, it is customary to provide sufficient data on how you support your findings and conclusions.
The figures are inadequate, and do not qualify to the standards of the journal. Using Wikipedia-based figures without even proper referencing in a scientific article is not a good idea. Also, referencing news articles from media organizations is not sufficient enough to support statistical data.
Importantly, the authors did not distinguish which age group they are analyzing. The manuscript uses “children” and “youth” terms interchangeably, which of course is not accurate, especially given the topic in question.
Additional comments:
- Line 89: “Moreover, the 2 million refugee children were widely scattered across receiving nations and there was no means for systematically tracking them or assessing their health status.”
- What is the age group in question? also I am sure that all migrants into neighboring European countries were properly documented before crossing the border
- Line 105 onwards: Screening for SARS-CoV-2 in children under 12 years is very infrequent, compared to screening for teenagers and adults. Additionally, vaccination is naturally lower in this age group, hence, the situation the authors are talking about is not only in Ukraine, as a result of war, but it is a given fact worldwide.
- Line 208: How does the war effect and migration to neighboring European countries increase the risk for HIV infection in children? Also how can the disruption of the Ukrainian Health system
All in all, unfortunately I cannot endorse the manuscript in its current form, and I hope the comments provided to the authors may aid in modifying the manuscript to reach journal standards.
Author Response
Reviewer #2
Note to reviewer: This paper was developed in response to a direct invitation from guest editor, Dr. Bozzola, to first author, Dr. Andrea Maggioni, to contribute a paper for the special IJERPH issue on preventable infectious diseases in children. The Ukraine conflict provided a compelling tale to tell, and we directed our writing to this evolving situation that threathens the health and welfare of Ukrainian children, with primary emphasis on infectious diseases.
Thank you for your review.
Comment: I have read the article by Maggioni et.al. with great interest. However, after thorough reading, I could not understand the aim of the manuscript. The authors provide commentary on the current situation in Ukraine, and how the recent war situation may lead to public health crisis in displaced children, in terms of infectious diseases, ultimately affecting the countries to which the migrant children may arrive to.
My biggest concern regarding this manuscript is that it does not qualify as an article, perhaps it is wiser to shorten the manuscript and use as a commentary or letter to the editor if feasible.
Response: The recommendation to change the paper to Commentary format is extremely helpful. We were not originally aware of this option. The Commentary style provides a much better fit for our paper. Thank you very much for this suggestion.
We changed the manuscript type to “Commentary” (line 1).
Comment: The manuscript is vague, does not provide sufficient evidence nor provide thorough discussion of the topic in question. Most importantly, the statistical data are not supported with proper referencing, and while the conclusions arrived to by the authors are logical and may in fact be the case, it is customary to provide sufficient data on how you support your findings and conclusions.
Response: We make 3 points in response. First, we have converted the paper to a Commentary, so the expectation that we produce a data-based paper is lessened. Second, this is written early in the conflict, so whatever data may eventually become available are not currently at hand. In fact, we mention that the war has severely disrupted infectious disease surveillance systems.
Comment: The figures are inadequate, and do not qualify to the standards of the journal. Using Wikipedia-based figures without even proper referencing in a scientific article is not a good idea. Also, referencing news articles from media organizations is not sufficient enough to support statistical data. Importantly, the authors did not distinguish which age group they are analyzing. The manuscript uses “children” and “youth” terms interchangeably, which of course is not accurate, especially given the topic in question.
Response: Independent from this comment, we had determined not to include the figures in a commentary style paper.
Comment: Line 89: “Moreover, the 2 million refugee children were widely scattered across receiving nations and there was no means for systematically tracking them or assessing their health status.”
What is the age group in question? also I am sure that all migrants into neighboring European countries were properly documented before crossing the border
Response: The age group is 0-17 years (now noted in the narrative). The data has not been made available to the public by hosting European countries. We mentioned in the article the information released by UNICEF, unfortunately, they have not shared data about specific age groups.
Comment: Line 105 onwards: Screening for SARS-CoV-2 in children under 12 years is very infrequent, compared to screening for teenagers and adults. Additionally, vaccination is naturally lower in this age group, hence, the situation the authors are talking about is not only in Ukraine, as a result of war, but it is a given fact worldwide.
Response: This is a valid critique. We have mentioned that testing in children is inaccurate, not only due to displacement, but also because they often develop mild disease and are not tested at all. We added a sentence to clarify this statement.
Comment: Line 208: How does the war effect and migration to neighboring European countries increase the risk for HIV infection in children? Also, how can the disruption of the Ukrainian Health system [note: this sentence was incomplete]
Response: We added several sentences in that paragraph to explain that displaced children and orphans are vulnerable to human trafficking and sexual abuse, which ultimately puts them at risk of sexual transmitted infections and HIV.

Reviewer 3 Report
Thank you for the invitation to review this paper. I have included the following comments for the authors to address:
- in the number 5 section (HIV risks in Ukrainian children and adolescents), the first has a number of typo errors - "Ukraine ranked second in Eastern Europe in HIV prevalence with an estimated 260,000 cases and 3,000 to 5,000 deaths in adults and adolescents annually [8]. Both intra-venous drug use and heterosexual transmission were prominent risks as HIV cases rose almost 10% annually from 2007 to 2017. Only 40% of HIV positive adults and 54% of HIV positive children were receiving anti-retroviral treatment before the [8]. Anti-retroviral treatment was strongly recommended to prevent vertical transmission from infected mothers to children. During the first months of the war, risks for HIV infection were sharply elevated for children on the move, including many who were unaccompanied, and for the nation’s 100,000 children, including orphans, who were living in institutional care facilities."
- As the authors discussed HIV risks in Ukrainian and have highlighted how access to healthcare was restricted by the Russian invasion, I suggest they include a discussion on restricted access to PMTCT among HIV-positive pregnant women which could give rise to HIV risks in Ukrainian children. The authors may also want to consider citing a recent paper on barriers PMTCT services uptake among HIV-positive women and how health systemic factors contribute: Ogueji, I. A., & Omotoso, E. B. (2021). Barriers to PMTCT services uptake among pregnant women living with HIV: A qualitative study. Journal of HIV/AIDS & Social Services, 20(2), 115-127. https://doi.org/10.1080/15381501.2021.1919276
- As the authors also discussed extensively about COVID-19 vaccination, supporting their arguments with recent related papers will be beneficial to their study. For instance, see reference: Ogueji, I. A., & Okoloba, M. M. (2022). Underlying factors in the willingness to receive and barriers to receiving the COVID-19 vaccine among residents in the UK and Nigeria: a qualitative study. Current Psychology, 1-12. https://doi.org/10.1007/s12144-021-02498-6
- As the authors discuss the vulnerability of Ukrainian children to abuse and violence, the authors' manuscript will benefit from supporting it with recent topics around this. For instance, see: Asagba, R. B., Noibi, O. W., & Ogueji, I. A. (2021). Gender Differences in Children’s Exposure to Domestic Violence in Nigeria. Journal of Child & Adolescent Trauma, 1-4. https://doi.org/10.1007/s40653-021-00386-6
- It is not clear what leaders/governments of other countries (for instance, non-affected countries) can learn from Russia's invasion of Ukraine. The authors should please make this clear.
- This is a clear and well-written paper (although there are minor grammatical/typo errors in various parts of the paper). The authors have failed to clearly spell out what recommendations they are making for governments, practitioners, and researchers. The authors should please address these shortcomings in their manuscript.
- It is not clear what search strategy the authors employed to generate their literature and what design this study adopted. The authors should make very clear within their manuscript.
- I recommend the acceptance of the manuscript after the authors have sincerely revised it.
Author Response
Reviewer #3
Note to reviewer: This paper was developed in response to a direct invitation from guest editor, Dr. Bozzola, to first author, Dr. Andrea Maggioni, to contribute a paper for the special IJERPH issue on preventable infectious diseases in children. The Ukraine conflict provided a compelling tale to tell, and we directed our writing to this evolving situation that threathens the health and welfare of Ukrainian children, with primary emphasis on infectious diseases.
Comment: Thank you for the invitation to review this paper.
Response: Thank you for your review.
I have included the following comments for the authors to address:
In the number 5 section (HIV risks in Ukrainian children and adolescents), the first has a number of typo errors - "Ukraine ranked second in Eastern Europe in HIV prevalence with an estimated 260,000 cases and 3,000 to 5,000 deaths in adults and adolescents annually [8]. Both intra-venous drug use and heterosexual transmission were prominent risks as HIV cases rose almost 10% annually from 2007 to 2017. Only 40% of HIV positive adults and 54% of HIV positive children were receiving anti-retroviral treatment before the [8]. Anti-retroviral treatment was strongly recommended to prevent vertical transmission from infected mothers to children. During the first months of the war, risks for HIV infection were sharply elevated for children on the move, including many who were unaccompanied, and for the nation’s 100,000 children, including orphans, who were living in institutional care facilities."
Response: We corrected the typos and modified this paragraph in the manuscript (lines 206-214)
Comment: As the authors discussed HIV risks in Ukrainian and have highlighted how access to healthcare was restricted by the Russian invasion, I suggest they include a discussion on restricted access to PMTCT among HIV-positive pregnant women which could give rise to HIV risks in Ukrainian children. The authors may also want to consider citing a recent paper on barriers PMTCT services uptake among HIV-positive women and how health systemic factors contribute: Ogueji, I. A., & Omotoso, E. B. (2021). Barriers to PMTCT services uptake among pregnant women living with HIV: A qualitative study. Journal of HIV/AIDS & Social Services, 20(2), 115-127. https://doi.org/10.1080/15381501.2021.1919276
Response: We have added several sentences about PMTCT. We added the suggested reference. Thank you!
Comment: As the authors also discussed extensively about COVID-19 vaccination, supporting their arguments with recent related papers will be beneficial to their study. For instance, see reference: Ogueji, I. A., & Okoloba, M. M. (2022). Underlying factors in the willingness to receive and barriers to receiving the COVID-19 vaccine among residents in the UK and Nigeria: a qualitative study. Current Psychology, 1-12. https://doi.org/10.1007/s12144-021-02498-
Response: We have mentioned in the manuscript the potential barriers and factors associated with intention to vaccinate. We added the suggested reference. Thank you!
Comment: As the authors discuss the vulnerability of Ukrainian children to abuse and violence, the authors' manuscript will benefit from supporting it with recent topics around this. For instance, see: Asagba, R. B., Noibi, O. W., & Ogueji, I. A. (2021). Gender Differences in Children’s Exposure to Domestic Violence in Nigeria. Journal of Child & Adolescent Trauma, 1-4. https://doi.org/10.1007/s40653-021-00386-6
Response: We added a sentence about the vulnerability of children to abuse and neglect. We added the suggested reference. Thank you!
Comment: It is not clear what leaders/governments of other countries (for instance, non-affected countries) can learn from Russia's invasion of Ukraine. The authors should please make this clear.
Response: Thank you for this suggestion. We are considering a follow-on policy paper, but this is not the direction that we are taking for this commentary.
Comment: This is a clear and well-written paper (although there are minor grammatical/typo errors in various parts of the paper). The authors have failed to clearly spell out what recommendations they are making for governments, practitioners, and researchers. The authors should please address these shortcomings in their manuscript.
Response: Thank you for this suggestion. We are considering a follow-on policy paper, but this is not the direction that we are taking for this commentary.
We are publishing a series of papers on a range of medically vulnerable patient populations during the Ukraine invasion with leading health officials in Ukraine. Thusfar, we have published papers on cancer care and kidney care, and a paper on eye care/ocular trauma has been revised and is under review. We intend to consolidate our papers into an overarching paper on medically vulnerable patient populations collectively. That paper will include some of the suggested recommendations.
Comment: It is not clear what search strategy the authors employed to generate their literature and what design this study adopted. The authors should make very clear within their manuscript.
Response: The paper is now formatted as a “commentary”. We did not use a standardized search strategy for this type of paper at this early stage of the conflict. We hope to follow this paper with a systematic review once sufficient scientific literature has been produced.

Reviewer 4 Report
Dear Authors,
I appreciate your effort in doing this study. The review is potentially good, but the representation of the study does not reflect that at all. Though it is an urgently required review article to be considered due to its importance under the current circumstances, it needs few important improvements before it should be considered. Please see my further comments below.
The different sections should be better formalized to improve the quality of reading.
Authors must incorporate the rationale of the study as an additional paragraph which will provide the structure to the review. There is no ending statement; that why this work is important? and what this review will provide as a novel information etc.? What methodology authors have used to collect the data and do the literature study? Authors should include a section on the bibliometric information for repeatability of the study.
I can see that the authors are native English speakers, but I strongly recommend authors to revise the manuscript for English language. There are huge amount of tense, singular plural, typo, grammatical etc errors. For example, at some places authors have mentioned infectious disease and some infectious diseases. All microbial names are written in non-Italics. Authors must understand the importance and ruling of writing the microbial name. Please see below further comments.
Abstract line 36-38 and Introduction line 54-57 are similar. Please revise.
Line 174-175: "the two-dose measles vaccination rate was just 81.9%". What do you mean by just? 82% is a good vaccination rate. Please revise the sentence.
Line 174-194: 20 lines are without references and stating bold claims. Authors must add official and state provided data references for these statements.
Line 187-188: "Ukraine have one of the highest rates of tuberculosis in the world 187 (73 incident cases per 100,000)".
It is a wrong statement. There are tens of many other countries above Ukraine in the list of rate of tuberculosis. However, yes it is in the list of MDR and RR TB list. So revise the sentence accordingly.
Line 206: "positive children were receiving anti-retroviral treatment before the........." Sentence incomplete.
Line 231-235: "Rape has been used a weapon of war to systematically terrorize and displace populations. Incidents of gender-based violence (GBV) perpetrated on Ukrainian women and youth at the hands of invading troops began to be reported within the first weeks of the invasion, with incidents proliferating during subsequent months of conflict and occupation....."
This is a bold claim without a proven or sufficient evidence. Even the reference authors have claimed for this sentence also states "allegations". So , choose the word very carefully. Authors must provide relevant references for bold claims.
Line 253, 261: Staphylococcus epidermidis and other microbial names- Italics
Line 265: gram.....it should be Gram... it's a name.
Section 8: Resources and challenges: Authors have focussed only on challenges but in the whole manuscript authors have not discussed about the resources (current facilities, emergency services, shelters, temporary housings etc) provided by the Government or any other agencies. In this section, also provide this data.
Authors contributions, conflict statement etc are missing.
When I started reading the review, i felt that authors have focussed more on COVID and there is several repetitions about that. Please reduce the COVID repetitive matter from the review. OR arrange a separate section in the review where discuss only about situation and COVID data. Do not mix with rabies, seasonal influenza, HIV etc. It is obstructing the readability.
Authors have repeated multiple times and used same words at many places regarding the Russian bombardment, war, Russian attack, destroyed, assault etc. I believe it is a scientific article and not a technical report. Please reduce the usage of these terms from the manuscript.
Author Response
Reviewer #4
Note to reviewer: This paper was developed in response to a direct invitation from guest editor, Dr. Bozzola, to first author, Dr. Andrea Maggioni, to contribute a paper for the special IJERPH issue on preventable infectious diseases in children. The Ukraine conflict provided a compelling tale to tell, and we directed our writing to this evolving situation that threathens the health and welfare of Ukrainian children, with primary emphasis on infectious diseases.
Comment: The different sections should be better formalized to improve the quality of reading.
Authors must incorporate the rationale of the study as an additional paragraph which will provide the structure to the review. There is no ending statement; that why this work is important? and what this review will provide as a novel information etc.? What methodology authors have used to collect the data and do the literature study? Authors should include a section on the bibliometric information for repeatability of the study.
Response: We have converted our paper to a Commentary. The abstract and introduction address the rationale for writing our invited paper on this topic. We did not conduct a formal study. As we were writing, the conflict was still in the early months and it was not possible to perform a formal literature review; almost no papers have yet been published. We searched ReliefWeb and related websites that collect and process information on evolving humanitarian crises.
Comment: I can see that the authors are native English speakers, but I strongly recommend authors to revise the manuscript for English language. There are huge amount of tense, singular plural, typo, grammatical etc errors. For example, at some places authors have mentioned infectious disease and some infectious diseases. All microbial names are written in non-Italics. Authors must understand the importance and ruling of writing the microbial name. Please see below further comments.
Response: The typos and grammatical errors were corrected. Microbial names are in italics in the manuscript now.
Comment: Abstract line 36-38 and Introduction line 54-57 are similar. Please revise.
Response: The abstract has been substantially rewritten, as has the introduction. There are some similarities in the information provided in both abstract and introduction. That is intentional, knowing that some readers will only read the abstract, while others will begin reading the narrative without having read the abstract. We are “framing” our narrative in both places.
Comment: Line 174-175: "the two-dose measles vaccination rate was just 81.9%". What do you mean by just? 82% is a good vaccination rate. Please revise the sentence.
Response: 82% is not a “good vaccination rate” for measles. We have added the following phrasing to explain:
Nevertheless, in February 2022, when the fighting broke out, the two-dose measles vaccination rate was just 81.9%. This percentage exceeds vaccination rates achieved for COVID-19, but this is deceptive. This level of vaccination is inadequate to protect against rapid outbreak spread, given measles’ very high reproduction number (the average number of persons who will be infected by one person with disease) and ease of transmissibility. This has produced a high-risk situation, particularly with most Ukrainian children on the move.
Comment: Line 174-194: 20 lines are without references and stating bold claims. Authors must add official and state provided data references for these statements.
Response:
We added the references in the manuscript. The data about vaccines comes from:
Ukraine: Public Health Situation Analysis (PHSA), 3 March 2022. Available at: https://www.humanitarianresponse.info/sites/www.humanitarianresponse.info/files/documents/files/ukraine-phsa-shortform-030322.pdf. Accessed June 1, 2022.
In terms of the information about Ukrainians traveling with their pets, we cited the reference in the manuscript too.
Comment: Line 187-188: "Ukraine have one of the highest rates of tuberculosis in the world 187 (73 incident cases per 100,000)". It is a wrong statement. There are tens of many other countries above Ukraine in the list of rate of tuberculosis. However, yes it is in the list of MDR and RR TB list. So revise the sentence accordingly.
Response: We revised the paragraph to read:
Tuberculosis (TB) remains a critical public health concern in Ukraine, particularly because the rate of multi-drug resistant tuberculosis (MDR-TB) is exceptionally high, accounting for 24-29% of newly diagnosed cases. Furthermore, Ukraine ranks second among European nations in HIV/TB coinfection; just under one-quarter (22%) of Ukrainians living with HIV are coinfected with TB.
Comment: Line 206: "positive children were receiving anti-retroviral treatment before the........." Sentence incomplete.
Response: We corrected the sentence:
Only 40% of HIV positive adults and 54% of HIV positive children were receiving anti-retroviral treatment before the war [8].
Comment: Line 231-235: "Rape has been used a weapon of war to systematically terrorize and displace populations. Incidents of gender-based violence (GBV) perpetrated on Ukrainian women and youth at the hands of invading troops began to be reported within the first weeks of the invasion, with incidents proliferating during subsequent months of conflict and occupation....."
This is a bold claim without a proven or sufficient evidence. Even the reference authors have claimed for this sentence also states "allegations". So , choose the word very carefully. Authors must provide relevant references for bold claims.
Response: We modified that statement and included the words “presumptive” and “allegations”, given the lack of proven evidence:
Rape has been used a weapon of war to systematically terrorize and displace populations. Presumptive incidents of gender-based violence (GBV) perpetrated on Ukrainian women and youth at the hands of invading troops began to be reported within the first weeks of the invasion, with allegations proliferating during subsequent months of conflict and occupation [24].
Comment: Line 253, 261: Staphylococcus epidermidis and other microbial names- Italics
Response: This has been corrected throughout.
Comment: Line 265: gram.....it should be Gram... it's a name.
Response: This has been corrected.
Comment: Section 8: Resources and challenges: Authors have focused only on challenges but in the whole manuscript authors have not discussed about the resources (current facilities, emergency services, shelters, temporary housings etc) provided by the Government or any other agencies. In this section, also provide this data.
Response:
Comment: Authors contributions, conflict statement etc are missing.
Response: Added:
Author Contributions: Conceptualization, A.M.1, J.M.S., J.A.G.-Z., A.M.2; literature review, J.A.G.-Z., J.M.S., S.A.M., A.M.2; writing—original draft preparation, J.A.G.-Z., J.M.S., L.P., U. von B., M.E., Z.E.; writing—review and editing, J.A.G.-Z, J.M.S.; supervision, A.M.1, J.M.S., J.A.G.-Z. All authors have read and agreed to the published version of the manuscript.
Funding: This collaborative writing project received no external funding.
Institutional Review Board Statement: Not applicable. No research study was conducted involving human subjects.
Informed Consent Statement: Not applicable. No research study was conducted involving human subjects.
Data Availability Statement: Not applicable.
Acknowledgments: None.
Conflicts of Interest: The authors declare no conflict of interest.
Comment: When I started reading the review, i felt that authors have focused more on COVID and there is several repetitions about that. Please reduce the COVID repetitive matter from the review. OR arrange a separate section in the review where discuss only about situation and COVID data. Do not mix with rabies, seasonal influenza, HIV etc. It is obstructing the readability.
Response: We have streamlined and reduced the writing that focuses on COVID-19.
Comment: Authors have repeated multiple times and used same words at many places regarding the Russian bombardment, war, Russian attack, destroyed, assault etc. I believe it is a scientific article and not a technical report. Please reduce the usage of these terms from the manuscript.
Response: This is now formatted as a Commentary. We have successfully published papers on cancer care, kidney care, and eye care/ocular trauma during the Ukraine conflict with comparable use of terminology in journals with strong impact factors. We believe our descriptions of the conflict are important for providing the context for the public health consequences broadly and for preventable infectious disease risks specifically.

Round 2
Reviewer 2 Report
The authors have clarified the concerning issues raised previously, and have modified their manuscript according to the suggestions. As a commentary/opinion article, I feel that this is an important contribution to the field and journal.
Author Response
Thank you very much for your helpful comments that have been very important in our selection of type of article (commentary) and have strengthened our paper.
Reviewer 3 Report
Congratulations to the authors for this excellent commentary! It is suitable for publication.
Author Response
Thank you very much for your helpful comments that have strengthened our paper.
Reviewer 4 Report
Manuscript is significantly improved by the authors. However, there is still one comment which authors have not responded to in the rebuttal. Please address this comment and resubmit accordingly.
“Comment (round 1): Section 8: Resources and challenges: Authors have focused only on challenges but in the whole manuscript authors have not discussed about the resources (current facilities, emergency services, shelters, temporary housings etc) provided by the Ukrainian government or any other agencies. In this section, also provide this data.”
Author Response
Reviewer #4
Comment: Section 8: Resources and challenges: Authors have focused only on challenges but in the whole manuscript authors have not discussed about the resources (current facilities, emergency services, shelters, temporary housings etc) provided by the Government or any other agencies. In this section, also provide this data.
Response: Thank you for requesting this important addition.
First, we have now expanded Section 8 to include the following narrative and 5 additional references:
8. Challenges and resources for monitoring communicable disease risks for Ukrainian refugees
Providing consistent and comprehensive infectious disease screening, timely intervention, and medical care for Ukrainian child refugees has been challenging due to many compounding factors [41]. The scale of forced migration, with large numbers of refugees and IDPs on the move, settling temporarily and relocating to other communities or countries, creates formidable obstacles for providing continuity of care as well as fortracking and maintaining accurate medical records.
Another complication involves refugee families arriving in European countries that have been hosted in private homes. Private residences are outside the jurisdiction of community public health services that focus on refugees in shelters and public housing. Refugees who are accommodated in private homes may not receive screening for infectious diseases and other critical health services. Going forward, it will be critical to integrate refugee children and their caregivers into community pediatric services in a timely and holistic manner.
Challenges notwithstanding, the Ukrainian refugee crisis has triggered an astonishingly rapid, rich, and resourced response. Since the beginning of the war, neighboring European countries such as Poland, Romania, Hungary, Slovakia, the Republic of Moldova, and Lithuania have generously provided a safe haven for the massive refugee flows.
Arriving refugees have been predominantly women, children, and caregivers with male family members remaining in Ukraine to defend the homeland. Refugees arrive with urgent needs for safety, security, and basic survival necessities. Border countries have opened a network of reception centers to provide migrants with shelter, food, health care, information, training, specialized interventions, and other essential services [42].
Children with complex medical conditions have been prioritized to receive care. Special attention has been given to continuing care for child cancer patients, including those with hematological conditions that may be complicated by resistant infection.
A broad spectrum of humanitarian actors, including many WHO and UN agencies, are on-scene, providing services both inside Ukraine and throughout European countries. Most pertinent for our focus on children is the work of UNICEF, playing a crucial role in providing support for refugee children and their families [43]. Inside Ukraine, UNICEF has distributed life-saving health and medical supplies for nearly 2.1 million people in war-affected areas, reaching more than 610,000 children and caregivers with mental health and psychosocial support services, and providing educational supplies to nearly 290,000 children.
In refugee-hosting countries, UNICEF is supporting national, municipal, and local systems that deliver essential services and protection, particularly for the most vulnerable children. This includes anti-trafficking training for border guards; expanding learning opportunities and integrating refugee children into schools; procuring vaccines and medical supplies; and establishing play and learning hubs that provide young children with a much-needed sense of normalcy and respite [43]. Almost 300,000 vulnerable families have registered for the UNICEF-Ministry of Social Policy humanitarian cash assistance program [43].
UNICEF has also established Blue Dots—safe spaces situated along border crossings in border countries that provide refugee children and families with critical information and practical support to help them in their onward journeys [44]. Blue Dot staff identify and register children travelling on their own and connect them to protection services. Blue Dots offer referral services to women, including those who are survivors of GBV. For children, Blue Dot hubs provide a safe, welcoming space to rest, play, and benefit from structured activities and psychosocial support from trained staff [44]. To date, 25 Blue Dots have been established along major transit routes in Moldova, Romania, Poland, Italy, Bulgaria, and Slovakia [43,44].
At border crossings and transit sites, medical care for refugees is being provided by both local humanitarian groups and international organizations, such as the Red Cross [45]. For example, in Poland, there are as many as 450 Red Cross medics working at border sites [45]. Provision of basic first aid and psychosocial support is supplemented by referrals to higher levels of care provided by national healthcare systems as necessary.
Multiple European governments have introduced legislation to allow refugees to access health care in hosting countries. On March 3, 2022, European Union (EU) members agreed to activate the Temporary Protection Directive for Ukrainian refugees, granting them access to health care across the EU. At least 10,000 hospital beds across the EU have been secured for refugee patients.
Vaccination centers have been established in receiving countries to ensure continuity of COVID-19 vaccinations and child immunizations for refugees. In the UK, a scheme has been devised to allow Ukrainian refugees to receive consultations and care through the National Health Service (NHS) [45].
Specific to infectious diseases, inside Ukraine, the Ministry of Health is working with local clinics and NGOS such as the Alliance for Public Health and 100% Life to facilitate availability of antiretroviral therapies for people living with HIV who have being displaced from their usual sources of care. The Ukrainian government has activated hotlines to provide guidance to people living with HIV, TB, and viral hepatitis. HIV care for refugees has also been offered at no cost by several European nations including Belgium, Bulgaria, Croatia, and Cyprus [46].
European national and international networks of pediatric infectious disease specialists have been fortified during the COVID-19 pandemic, along with clinical trials networks for pediatric TB and HIV. During the war, these established networks have facilitated close contacts with Ukrainian counterparts. Among the medical societies active in this process, the European Society for Paediatric Infectious Diseases (ESPID) and the European Academy of Paediatrics (EAP) have forged close linkages with the Royal College of Paediatrics and Child Health (RCPCH) in the UK to share critical health information regarding preventable infectious diseases and other health conditions for Ukrainian refugee children and adolescents [41,47-50].
Second, the concluding comment also has been expanded to include this note:
Concerted efforts on the part of networks of infectious disease experts and pediatric clinicians have attempted to intervene to optimize health and prevent infectious disease spread among this vulnerable population of conflict-displaced Ukrainian youth. For example, The Ukraine Children’s Action Project was established to support displaced Ukrainian children’s health and educational needs. As the conflict rages on, more children will be injured, displaced, and left at risk to infectious disease spread. These efforts are therefore vital for protecting the health and well-being of the youngest survivors of the war.
